# Three-Color White Photoluminescence Emission Using Perovskite Nanoplatelets and Organic Emitter

**DOI:** 10.3390/molecules27133982

**Published:** 2022-06-21

**Authors:** Hyukmin Kwon, Sunwoo Park, Seokwoo Kang, Hayoon Lee, Jongwook Park

**Affiliations:** Integrated Engineering, Department of Chemical Engineering, Kyung Hee University, Yongin 17104, Korea; hm531@khu.ac.kr (H.K.); mdrafix@khu.ac.kr (S.P.); swkang@khu.ac.kr (S.K.); kssarang1@khu.ac.kr (H.L.)

**Keywords:** aggregation-induced emission, metal halide perovskite, tetraphenylethylene, three-color white emission

## Abstract

Three organic blue-light-emitting tetraphenylethylene (TPE) derivatives that exhibit aggregation-induced emission (AIE) were used as additives in the preparation of inorganic perovskite-structured green-light-emitting materials for three-color white-light emission. For these organic–inorganic light-emitting materials, two-color (blue and green) light-emitting films based on the CsPbBr_3_ perovskite-structured green-light-emitting inorganic material were prepared. The three TPE derivatives were prepared by varying the number of bromide groups, and a distinct AIE effect was confirmed when the derivatives were dissolved in a water–tetrahydrofuran mixed solvent containing 90 vol% water. When 0.2 molar ratio of the 1,1,2,2-tetrakis(4-bromophenyl)ethylene (TeBrTPE) additive was mixed with nanocrystal-pinning toluene solvent, the green-light-emission photoluminescence quantum efficiency (PLQY) value at 535 nm was 47 times greater than that of the pure bulk CsPbBr_3_ without additives and a blue emission at 475 nm was observed from the TeBrTPE itself. When a CBP:Ir(piq)_3_ film was prepared on top of this layer, three PL peaks with maximum wavelength values of 470, 535, and 613 nm were confirmed. The film exhibited white-light emission with CIE color coordinates of (0.25, 0.36).

## 1. Introduction

Organic light-emitting diode (OLED) technology has grown into a representative display technology over the past 30 years because of its advantages of high electroluminescence (EL) efficiency, high resolution, and low energy consumption and because it can be applied to this and light flexible displays [1,2,3,4]. However, the high material cost and low color purity of organic emitters used in OLEDs are major obstacles to product development and to the realization of devices that display vivid and natural colors. To address these problems, researchers have widely investigated AMX_3_-based metal halide perovskites (MHPs) for use in light-emitting diodes (LEDs), photodetectors, and photovoltaic applications because of their narrow full-width at half-maximum (FWHM) emission band, excellent color purity and charge carrier mobility, tunable optical bandgap, and easy preparation via solution processes [5,6,7,8,9,10,11]. Here, *A* is an organic or inorganic cation such as methylammonium (MA^+^), formamidinium (FA^+^), or Cs^+^; *M* is a divalent metal cation such as Pb^2+^ or Sn^2+^; and *X* is a halide such as Cl^−^, Br^−^, or I^−^. However, defect sites on the surface of MHPs adversely affect both their optical and electrical properties. Consequently, researchers have passivated the defect sites of nanocrystals with specific materials such as organic ligands, inorganic species, or polymers to enhance the photoluminescence (PL) intensity of MHP materials and improve the operating stability of perovskite-based LEDs [12,13]. Representative organic ligands are MA, FA, and phenylethylammonium (PEA^+^) which are small organic-salt-type ligands [14,15,16]. In addition, small metal cations used for passivation, such as Na^+^ ions, have a smaller ionic radius than the metal ions in the perovskite component, enabling the small metal cations to occupy interstitial sites or lattice vacancies and easily passivate the defect sites [17].

Unlike previously reported studies in which perovskite materials were prepared with ammonium derivatives, the present work aims to improve the luminescence properties of perovskite materials by incorporating light-emitting organic compounds that exhibit aggregation-induced emission (AIE). Organic emitter derivatives based on anthracene, pyrene, or perylene chromophores, which are generally used for OLEDs, exhibit high emission efficiency in dilute solutions but weak emission in the aggregated film state because of the aggregation-caused quenching effect [18,19,20]. However, some new luminescent materials have an opposite effect, demonstrating bright-light emission upon aggregation in the film state. Typically, tetraphenylethylene (TPE) derivatives have a propeller-shaped structure that is a representative chemical structure exhibiting AIE characteristics [21,22,23,24,25,26,27]. Therefore, in the present study, we focus on new additives for perovskite-based light-emitting material components by using TPE and Br-substituted TPE derivatives to improve the PL light-emitting performance of film-state perovskites.

White LEDs (WLEDs) are important for lighting and display applications; however, perovskite-based WLEDs have rarely been reported. Recently, perovskite-based WLEDs with organic light-emitting materials have been reported to emit white light with two or three wavelengths through a single emitting layer or a double emitting layer [28,29,30,31,32,33]. One potential approach to achieving white-light electroluminescence through a single emitting layer is to use a perovskite emitter mixed with organic molecules such as small molecules or polymers. However, this method can realize only two-color white-light emission based on blue and yellow, not three-color white-light emission based on red, green, and blue. Three separate primary colors are basically required for a full-color display.

Because TPE derivatives emit in the blue region and CsPbBr_3_ emits in the green region, two-color emission of this mixed state can be applied as a component of three-color emission. In addition, if a red-emitting layer is placed on top of a two-color emissive layer, three-color white emission can be attained. We used this approach in the present study. Specifically, we used a solution process to fabricate an organic–inorganic mixed film capable of simultaneously emitting blue and green light and subsequently realized three-color white PL emission of a double layer by depositing tris(1-phenylisoquinoline)iridium(III) (Ir(piq)_3_), a red-light-emitting material, onto the two-color emissive layer. The results suggest that three-color white EL emission is feasible.

## 2. Experimental Section

### 2.1. Synthesis

#### 2.1.1. 1-Bromo-4-(1,2,2-triphenylethylene)benzene

Diphenylmethane (2.02 g, 12.00 mmol) was dissolved in 30 mL dry tetrahydrofuran (THF) and cooled to −5 °C in an ice bath. *n*-Butyllithium solution (6.25 mL, 5.50 mmol, 1.6 M in *n*-hexane) was added dropwise to the diphenylmethane solution, and the resultant mixture was stirred at −5 °C for 30 min. 4-Bromobenzophenone (2.35 g, 9.00 mmol) was added under a N_2_ atmosphere, the ice bath was removed, and the reaction mixture was warmed to room temperature and stirred at this temperature for 6 h. Afterwards, 50 mL of a saturated ammonium chloride solution was added and extracted with a dichloromethane–distilled water solution. The organic layer was dried over anhydrous MgSO_4_, and the solution was concentrated under reduced pressure. The crude product was used without further purification during the following reaction step and was placed under a N_2_ atmosphere and dissolved in 80 mL of toluene; *para*-toluene sulfonic acid (0.342 g, 1.8 mmol) was then added. The reaction mixture was heated to 120 °C and stirred at this temperature for 12 h. When the reaction was completed, the crude product was extracted with CH_2_Cl_2_–distilled water. The organic layer was dried over anhydrous MgSO_4_, and the solution was concentrated under reduced pressure. After column purification using ethyl acetate:hexane = 1:100, reprecipitation from THF and methanol gave a white solid. (47% yield) ^1^H NMR (400 MHz, DMSO): δ (ppm) 7.30–7.28 (m, 2H), 7.15–7.05 (m, 9 H), 6.95–6.91 (m, 6 H), 6.87–6.85 (m, 2 H). ^13^C NMR (101 MHz, DMSO): δ 143.42, 143.37, 143.21, 143.01, 141.77, 139.84, 133.31, 131.35, 131.18, 131.16, 131.09, 128.55, 128.49, 128.39, 127.36, 127.29, 127.24, 120.30 ppm. HRMS (FAB-MS, *m*/*z*): calcd. for C_26_H_19_Br, 410.07; found, 410.412 [M]^+^.

#### 2.1.2. 1,1,2,2-Tetrakis(4-bromophenyl)ethylene (TeBrTPE)

Bromine (9.6 g, 60.00 mmol) was added to a solution of tetraphenylethylene (2.50 g, 7.50 mmol) in 15 mL of glacial acetic acid and 30 mL of methylene chloride at 0 °C. The resultant mixture was stirred at room temperature for 4 h, poured into 100 mL ice water and extracted with CH_2_Cl_2_. The organic product was washed with water, brine and then dried over MgSO_4_; the solvent was subsequently removed under reduced pressure. The crude product was purified by recrystallization from methanol to give 1,1,2,2-tetrakis(4-bromophenyl)ethylene (TeBrTPE) as a white solid. (92% yield) ^1^H NMR (400 MHz, DMSO): δ (ppm) 7.37–7.34 (m, 8H), 6.90–6.86 (m, 8 H). ^13^C NMR (101 MHz, DMSO): δ 141.99, 139.85, 133.29, 131.68, 120.98 ppm. HRMS (FAB-MS, *m*/*z*): calcd. for C_26_H_16_Br_4_, 643.80; found, 644.185 [M]^+^.

### 2.2. Materials and Instruments

Diphenylmethane, *n*-Butyllithium solution (1.6 M in *n*-hexane), 4-Bromobenzophenone, TPE, 9,10-dibromoanthracene (9,10-DBA), cesium bromide (CsBr), lead(II) bromide (PbBr_2_), anhydrous dimethyl sulfoxide (DMSO), and toluene were purchased from Sigma-Aldrich. Hydrobromic acid (HBr) solution (48% in water) was purchased from Alfa Aesar. All chemicals were used without further purification. Optical absorption spectra were recorded using a Hewlett-Packard 8453 ultraviolet–visible–near-infrared (UV–Vis–NIR) spectrophotometer. The ^1^H NMR spectra were recorded on Bruker Advance 400 spectrometers (Bruker, Billerica, MA, USA). The FAB ± mass was recorded on a JMS-700. A Perkin-Elmer LS55 luminescence spectrometer (Perkin-Elmer, Waltham, MA, USA, light source: 20 kw pulsed Xe flash tube, filter: 1%, slit width: 10:10 nm) was used to record PL spectra. Absolute photoluminescence quantum yield (PLQY) values were obtained using a Quantaurus-QY quantum yield spectrometer (Hamamatsu photonics, Hamamatsu, Japan). The time-resolved fluorescence data were obtained using a Hamamatsu Photonics C11367-11 fluorescence lifetime spectrometer (Hamamatsu photonics, Hamamatsu, Japan). The average lifetimes of the emission decay were obtained by the equation [34]. X-ray diffraction (XRD) data were obtained using a Bruker D8 Advance diffractometer (Bruker, Billerica, MA, USA). Scanning electron microscopy (SEM) images were obtained using a JEM-2100F (JEOL, Toyko, Japan) electron microscope operating at 200 kV. CIE color coordinates of the perovskite organic film were measured at room temperature and under ambient conditions using a Konica Minolta CS-100 spectroradiometer (Konica Minolta, Toyko, Japan).

### 2.3. Perovskite Solution Preparation

CsPbBr_3_ powder was prepared and used as one of the starting materials for film preparation. We dissolved PbBr_2_ (10 mmol, 3.67 g) in HBr (8 mL), to which we then slowly added CsBr (10 mmol, 2.12 g, dissolved in 3 mL of water), resulting in an orange precipitate. The precipitate was filtered, washed twice with ethanol, and dried at 60 °C in a vacuum oven for 12 h before use. The synthesized CsPbBr_3_ (0.3 mmol, 0.174 g) was dissolved in 1 mL of DMSO to prepare a perovskite solution. Nanocrystal-pinning (NCP) toluene solvent was then mixed with different amounts of TPE or its derivatives (i.e., BrTPE and TeBrTPE). For example, to prepare a non-solvent having a 1:1 CsPbBr_3_-to-TPE mole concentration ratio, a mass of 0.099 g (0.3 mmol) of TPE was dissolved in toluene.

### 2.4. Perovskite Films Preparation

Glass substrates (25 mm × 25 mm) were sequentially washed by ultrasonication in acetone, ethanol, deionized water, and isopropyl alcohol and then dried in an oven at 80 °C. The substrates were further cleaned with a UV–ozone cleaner for 10 min before the spin-coating process. A CsPbBr_3_ solution was spin-coated at 3000 rpm for a total of 60 s; at the midpoint of the spin-coating process (i.e., after 30 s), 0.2 mL of NCP toluene solvent including TPE or one of its derivatives was dropped onto the surface; the coated substrate was then heat-treated at 70 °C for 1 min.

## 3. Results and Discussion

### 3.1. AIE Characterization of TPE Derviatives

Among the numerous organic materials that can potentially be added to perovskite light-emitting materials, we used organic materials that exhibited AIE properties to achieve the three-wavelength PL phenomenon. As shown in Figure 1, three TPE derivatives were selected and synthesized in the present study because TPE-based compounds are easy to synthesize and because the halide substitution effect enables the change in luminescence properties depending on the AIE properties to be easily characterized. The synthesized compounds were characterized using NMR spectroscopy (Appendix A). Because TPE has a large bandgap such as 3.7 eV, it emits in the blue region (467 nm) in the solution state [35]. BrTPE and TeBrTPE show PL maxima of 482 and 490 nm, which are bathochromically shifted compared with that of TPE; however, these results show that the large bandgap and blue emission of TPE are retained even when one or more halide atoms are substituted into its structure.

To investigate the AIE characteristics of TPE, BrTPE, and TeBrTPE, we recorded the PL spectra of samples prepared using a mixture of THF as a good solvent and H_2_O as a non-solvent. Figure 2 and Appendix A shows the PL spectra of the TPE derivatives in a THF–water solvent with a water fraction (f_w_) of 90 vol%. In water fraction (f_w_) of 90 vol%, PL maximum peak of TPE is 465 nm. In the case of BrTPE, the emission peak of BrTPE was observed at 485 nm, which was redshifted by ~20 nm from that of TPE. In the case of TeBrTPE, the emission peak appeared at 490 nm, which was redshifted by ~25 nm from that of TPE. When we use PLQY measurement in solution state, it is very difficult to measure the clear value due to the low PL quantum yield.

### 3.2. Optical Properties of Organic–Inorganic CsPbBr_3_ Nanoplatelet Films with an Organic Emitter

Figure 3 shows the ultraviolet–visible absorption spectra of the CsPbBr_3_ films containing TPE derivatives at various molar ratios. The spectrum of the bulk CsPbBr_3_ shows broad absorption at wavelengths less than 530 nm, and the films with TPE-derivative/CsPbBr_3_ molar ratios of 0.2–0.6 show a TPE absorption peak at ~315 nm. Bulk perovskite has been reported to exhibit absorption in the 430–470 nm range when transformed into a 2D low-dimensional phase by the addition of organic ligands [11,36]. However, when TPE and the Br-substituted TPE derivatives were added to CsPbBr_3_, a new absorption peak at 430–470 nm was not observed. The non-normalized PL spectra and the related optical properties of the CsPbBr_3_ films with different molar ratios of CsPbBr_3_ and TPE derivatives are shown in Figure 4, Appendix A, and Table 1, respectively. The spectrum of the pure CsPbBr_3_ film shows a weak maximum PL peak at 535 nm. When the TPE/CsPbBr_3_ molar ratio was increased from 0.2 to 0.6, two emission peaks were confirmed: one at 446 nm, which was the emission peak of TPE, and one at 535 nm, which was the emission peak of CsPbBr_3_. In the case of BrTPE, the emission peak of BrTPE was observed at 471 nm, which was redshifted by ~5 nm from that of TPE. In the case of TeBrTPE, the emission peak appeared at 476 nm, which was redshifted by ~9 nm from that of TPE. These results are attributed to enhancements in both the interaction of the TPE compound and CsPbBr_3_ and the grain-boundary defect passivation of CsPbBr_3_ with increasing number of bromides and increasing amount of the TeBrTPE derivative. Figure 4d shows a plot of the absolute PLQY ratio (*I*/*I*_0_) as a function of the molar ratio of TPE derivatives.

Table 1 shows the PL properties of the CsPbBr_3_ films containing the TPE compounds. The absolute PLQY of all the samples was measured at 535 nm (i.e., the green emission region); the pure CsPbBr_3_ film showed a PLQY of 0.4%. When TPE was added at molar ratios of 0.2, 0.4, and 0.6, the PLQY values increased to 0.7%, 1.2%, and 1.2%, respectively; however, no substantial difference was observed in the PLQY values of the TPE additive cases. However, when the BrTPE additive was added at the same concentrations as the TPE additive, the PLQY was 5.9%, 9.6%, and 13.8%; in addition, we confirmed that the PLQY was improved approximately tenfold compared with that when TPE was added. Furthermore, when TeBrTPE was added to the CsPbBr_3_ film at the same molar ratios, high PLQY values of 12.2%, 16.2%, and 18.6% were attained, consistent with the PL intensities in Figure 4. These values are approximately 47 times greater than the PLQY values corresponding to pure CsPbBr_3_. The related data showed an error range of less than 5%. Thus, we confirmed that the luminous efficiency of CsPbBr_3_ was improved because of AIE, and that two-color (blue and green) emission was realized through optimization of the ratio between the TPE derivatives and the perovskite material.

### 3.3. Morphologies of Organic–Inorganic CsPbBr_3_ Nanoplatelet Films with an Organic Emitter

In general, low-dimensional perovskite materials of two-dimensional (2D) and colloidal nanocrystal types with a large exciton binding energy and good film morphology are used to improve the emission efficiency of perovskite materials. In bulk-type perovskite materials, the emission intensity is weak; however, with the addition of an organic ammonium salt to induce a nanoplatelet shape through dimensional engineering and reduce the grain size, the exciton-binding energy can be increased, and the emission efficiency can be improved [37,38,39]. In the case of perovskite nanocrystals, the emission intensity can be improved by converting a bulk perovskite material into its corresponding perovskite quantum-dot form using long-chain organic ligands such as oleic acid and oleylamine [40,41]. The TPE derivatives used in the present study have not been previously used as a passivation additive. The TPE derivatives even exhibit light-emitting properties as well as AIE properties. XRD and SEM analyses were conducted to investigate the effects of the TPE derivatives on the dimensions of the formed perovskite and the morphology of the corresponding perovskite films.

The XRD patterns of CsPbBr_3_ films containing TPE, BrTPE, and TeBrTPE at molar ratios from 0.2 to 0.6 are shown in Figure 5a–c, respectively. The pattern of the pure CsPbBr_3_ thin film shows peaks at 2θ values of 15.3°, 21.7°, and 30.1°, which correspond to the (100), (110), and (200) planes of the three-dimensional perovskite structure. Interestingly, as the molar ratio of the TPE derivatives was increased, the intensity of the (110) diffraction peak decreased and the intensities of the (100) and (200) diffraction peaks increased. In the patterns of the films containing TPE, the (100) and (200) diffraction peaks slightly increased compared with those of bulk CsPbBr_3_. However, when BrTPE and TeBrTPE were added, the diffraction peak intensities of the (100) and (200) planes clearly increased with increasing molar ratio of the TPE derivatives. In particular, the strongest diffraction peak intensities were observed for TeBrTPE with a TeBrTPE/CsPbBr_3_ molar ratio of 0.6. This result means that the bromide of the TPE derivatives contained in the CsPbBr_3_ film can be predominantly passivated on the (100) and (200) crystal planes of CsPbBr_3_ crystallites and promote the growth of CsPbBr_3_ nanoplatelets along the (100) direction as well as the stacked formation [42]. Therefore, with increasing content of the TPE derivatives, the crystallinity of CsPbBr_3_ tended to increase along with the PL intensity. In general, the XRD peak corresponding to dimensional control of CsPbBr_3_ appears at a 2θ angle of 5°. When the TPE derivative was added, the dimension-controlled XRD peak at 5° was not observed irrespective of the derivative’s molar ratio. The same result has been reported by Li and coworkers [42,43].

SEM was used to analyze the changes in the CsPbBr_3_ film morphology depending on the addition of TPE derivatives (Figure 5d). The pure CsPbBr_3_ film contains pinholes and large crystal domains. When TPE was added to CsPbBr_3_ at molar ratios of 0.2, 0.4, and 0.6, the pinholes and grain size decreased but the surface uniformity worsened. When BrTPE or TeBrTPE was added to CsPbBr_3_ at molar ratios of 0.2, 0.4, and 0.6, the number of pinholes decreased, and defect passivation was observed. This difference in SEM observations of CsPbBr_3_ containing BrTPE and TeBrTPE compared with those of the CsPbBr_3_ containing TPE is attributable to the aggregation effect of TPE derivatives as well as to the interaction between TPE derivatives and CsPbBr_3_.

As a result, the luminous efficiency of CsPbBr_3_ samples containing a TPE derivative could increase, which can be explained by three factors. The first factor is the AIE effect of the TPE compounds. The PLQY of CsPbBr_3_ films containing TPE derivatives was improved compared with that of a pure CsPbBr_3_ film (Table 1). 9,10-DBA, an anthracene derivative without AIE properties, was added to a CsPbBr_3_ film under the same conditions to investigate whether the AIE property of the TPE improved the luminous efficiency of the CsPbBr_3_ film. The PL properties of these films were investigated; the PL spectra are shown in Appendix A. Anthracene is a chromophore widely used in OLEDs because it has a high PLQY as a blue-light-emitting material. However, the results for films with various molar ratios of 9,10-DBA show that the PL intensity of CsPbBr_3_ did not increase; a PL intensity similar to that of bulk CsPbBr_3_ was observed. These results indicate that the AIE effect of the TPE derivatives enhanced the PL intensity of the CsPbBr_3_ films. The second factor is the improvement in the luminous efficiency of CsPbBr_3_ by the interaction of CsPbBr_3_ with the Br substituted onto TPE. We previously confirmed that the PL intensity and PLQY increased in the order TPE, BrTPE, and TeBrTPE. In addition, the decay time of the light intensity based on the PL mission at 535 nm confirmed that an interaction occurred between the CsPbBr_3_ and the TPE additives (Appendix A). The average decay time, *τ*_avg_, of CsPbBr_3_ was 0.85 ns. In the case of PL emission at 535 nm, the CsPbBr_3_ films containing TPE at molar ratios of 0.2, 0.4, and 0.6 exhibited *τ*_avg_ values of 1.78, 1.86, and 2.08 ns, respectively; the CsPbBr_3_ films containing BrTPE at molar ratios of 0.2, 0.4, and 0.6 exhibited *τ*_avg_ values of 4.04, 4.25, and 4.57 ns, respectively; and the CsPbBr_3_ films containing TeBrTPE at molar ratios of 0.2, 0.4, and 0.6 exhibited *τ*_avg_ values of 5.16, 5.29, and 5.55 ns, respectively. Thus, as the TPE derivative content was increased, the decay time increased, indicating that the TPE derivatives interacted with the perovskite material. That is, we inferred that the delay time was longer than the average decay time of the perovskite because of an interaction of the TPE derivatives and the perovskite, where energy is transferred from the large-bandgap TPE derivatives to the perovskite. The third factor is perovskite grain size control and defect passivation. SEM observations revealed that, with decreasing grain size, the number of pinholes decreased. When a perovskite material is used as an efficient light-emitting layer, it is necessary to improve the radiative recombination effect by controlling the crystal grain size of the perovskite. A small crystal grain size enables effective spatial confinement of the excitons, thereby inhibiting the dissociation and diffusion of excitons. Finally, small crystal grains can lead to a PLQY enhancement of the perovskite material [37]. Moreover, when an AIE organic compound is added, PLQY enhancement through defect passivation can be confirmed.

### 3.4. Implementation of Three-Color White PL Emission in a Double Emitting Layer

Because a perovskite film can be prepared by a solution process and exhibits excellent optical properties, white-light-emitting devices with perovskite and organic light-emitting materials are being actively developed. However, blue-light-emitting perovskite materials exhibit weak luminous efficiency and poor stability compared with red- and green-light-emitting perovskite materials [11,41]. In addition, the emission band of blue-light emitting perovskites has a narrow FWHM, making white-light emission difficult. However, TPE derivatives have stable light emission in the blue region and the FWHM of their emission band is wider than that of perovskite materials; thus, the use of perovskite and TPE materials together should be advantageous in realizing white-light emission.

To achieve ideal white-light emission on the basis of the results of the present study, we prepared a perovskite film with a TeBrTPE/CsPbBr_3_ molar ratio of 0.4, which had a high PLQY as well as 1:1 ratio of blue and green PL intensities. We then vacuum-deposited a red phosphorescent material, 4,4′-Bis(N-carbazolyl)-1,1′-biphenyl (CBP): 9 wt% Ir(piq)_3_, on top of the perovskite film with TeBrTPE (0.4 molar ratio) to achieve a thickness of 20 nm, resulting in a double emitting layer. This film exhibited three-color white PL emission with a CIE value of (0.25, 0.36) as shown in Figure 6. By applying an organic light-emitting ligand with an AIE moiety to a perovskite material, we improved the luminous efficiency of CsPbBr_3_, suggesting that a three-color white LED device was feasible. Further studies on a three-color white LED are underway.

## 4. Conclusions

By applying an organic light-emitting ligand with AIE characteristics to CsPbBr_3_ at various molar ratios, we improved the luminous efficiency of the perovskite light-emitting material and obtained a film that simultaneously emitted green and blue light. The role of the AIE effect and the number of substituted halides in the TPE derivatives was investigated to improve the luminescence efficiency as well as the passivation effect resulting from the small grain size and ligand interaction of the perovskite material. The CsPbBr_3_ films were prepared using a nanocrystal-pinning solution mixed with TPE derivatives at different molar ratios of 0.2 to 0.6. When TeBrTPE was added to the CsPbBr_3_ film at a molar ratio of 0.6, the PLQY value of CsPbBr_3_ was 18.6%, which was approximately 47 times higher than the PLQY value of pure CsPbBr_3_ (0.4%). In particular, when TeBrTPE was added, the crystallinity of the CsPbBr_3_ was substantially increased. We confirmed that the interaction of the perovskite crystal domain as well as the PL intensity of CsPbBr_3_ were increased. As a result, upon deposition of a red phosphorescent material, CBP:Ir(piq)_3_, onto a CsPbBr_3_ film with a TeBrTPE/CsPbBr_3_ molar ratio of 0.4, which exhibited high PL efficiency, three distinct wavelengths of 469, 535, and 613 nm in the double layer were observed. This double-layer film also exhibited three-color white-light PL emission with a CIE value of (0.25, 0.36).

## Figures and Tables

**Figure 1 molecules-27-03982-f001:**
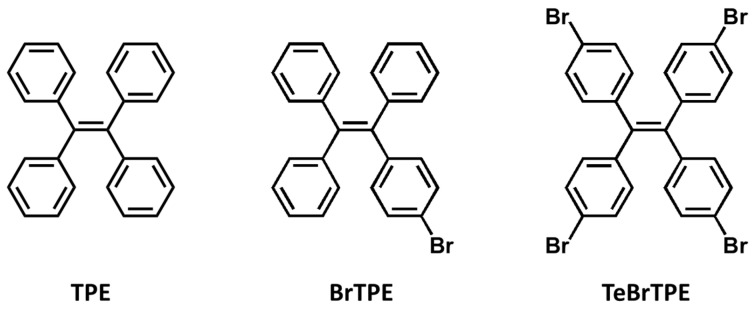
The chemical structures of the synthesized TPE derivatives used as organic spacers in CsPbBr3 films.

**Figure 2 molecules-27-03982-f002:**
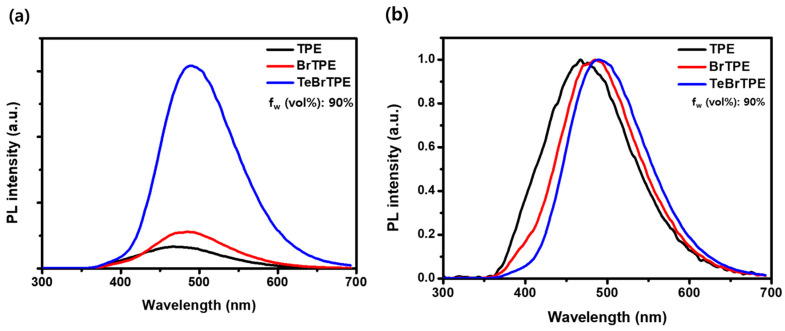
(**a**) Fluorescence spectra of TPE derivatives in THF (10 vol%) and water (90 vol%) solvents; (**b**) Normalized fluorescence spectra of TPE derivatives in THF (10 vol%) and water (90 vol%) solvents.

**Figure 3 molecules-27-03982-f003:**
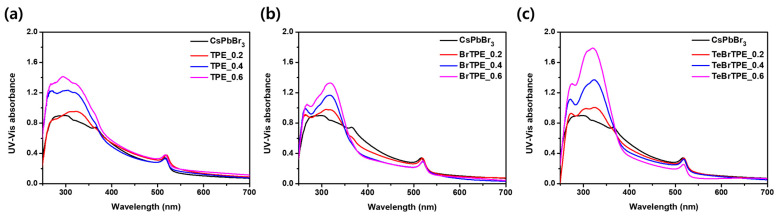
Absorption spectra of perovskite films with various molar ratios of TPE derivatives: (**a**) CsPbBr_3_ perovskite films with TPE/CsPbBr_3_ molar ratios of 0.2, 0.4, and 0.6; (**b**) CsPbBr_3_ perovskite films with BrTPE/CsPbBr_3_ molar ratios of 0.2, 0.4, and 0.6; and (**c**) CsPbBr_3_ perovskite films with TeBrTPE/CsPbBr_3_ molar ratios of 0.2, 0.4, and 0.6.

**Figure 4 molecules-27-03982-f004:**
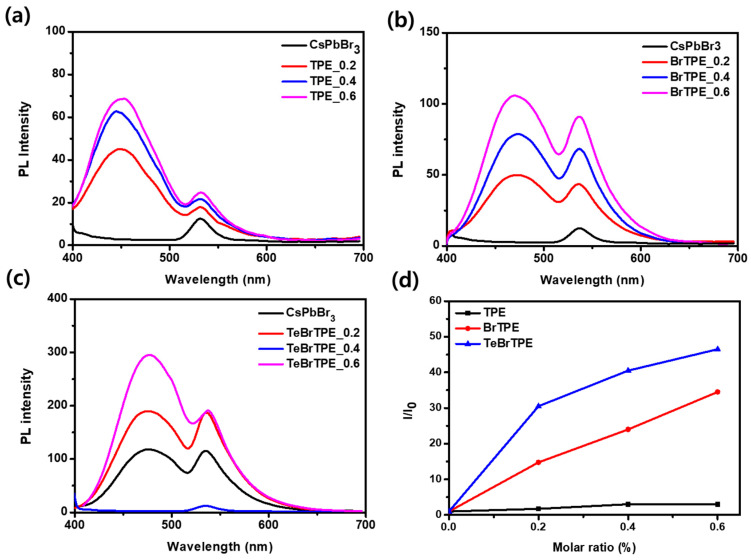
Photoluminescence (PL) spectra of CsPbBr_3_ films containing various TPE derivatives: (**a**) CsPbBr_3_ perovskite films with TPE/CsPbBr_3_ molar ratios of 0.2, 0.4, and 0.6; (**b**) CsPbBr_3_ perovskite films with BrTPE/CsPbBr_3_ molar ratios of 0.2, 0.4, and 0.6; and (**c**) CsPbBr_3_ perovskite films with TeBrTPE/CsPbBr_3_ molar ratios of 0.2, 0.4, and 0.6. (**d**) Plot of the relative emission intensity (*I*/*I*_0_) as a function of the molar ratio of TPE derivatives, where *I* is the absolute PLQY value at 535 nm and *I*_0_ is the absolute PLQY value at 535 nm in the spectrum of a pure CsPbBr_3_ film (excitation wavelength: 365 nm).

**Figure 5 molecules-27-03982-f005:**
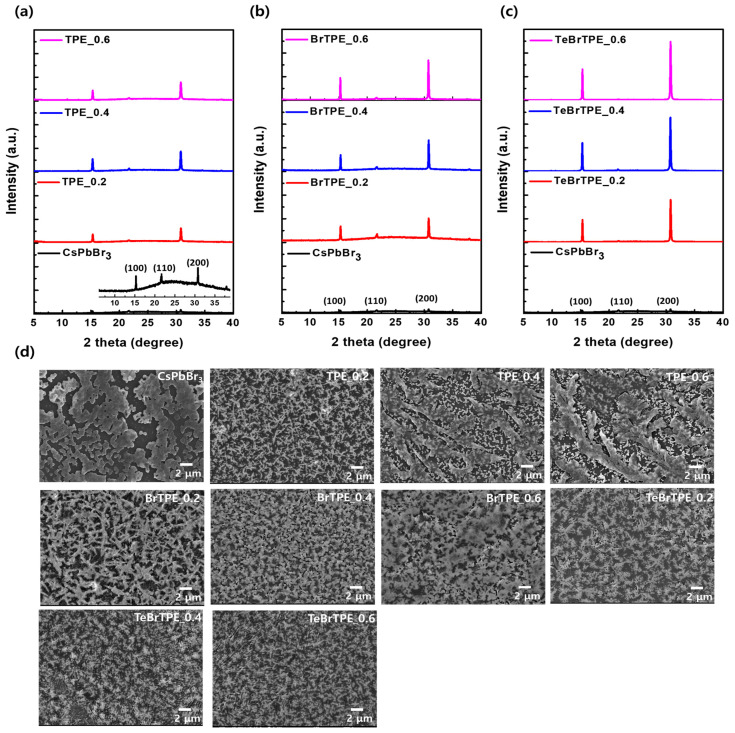
Crystal structures and morphologies of CsPbBr_3_ thin films with 0.2, 0.4, and 0.6 mol of TPE, BrTPE, and TeBrTPE per mole of CsPbBr_3_. (**a**–**c**) X-ray diffraction (XRD) patterns and (**d**) top-view scanning electron microscopy (SEM) images (25,000×, scale bar: 2 μm).

**Figure 6 molecules-27-03982-f006:**
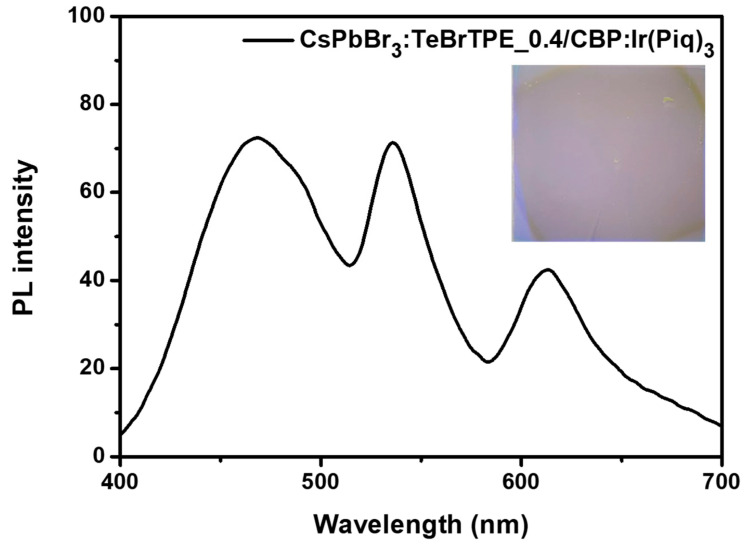
White-light PL spectrum of a vacuum-deposited film of a red emitter on a CsPbBr_3_ film with a BrTPE/CsPbBr_3_ molar ratio of 0.4 (excitation wavelength: 300 nm), inset: optical photograph under irradiation of 365 nm.

**Table 1 molecules-27-03982-t001:** Summary of the PL Peak Wavelength, PLQY, and *τ*_avg_ of CsPbBr_3_ Including Various Relative Molar Amounts of TPE Compounds in the Film State.

CsPbBr_3_	TPE Compound and Its Molar Ratio	PL Peak(s) (nm)	PLQY (%) (at 535 nm)	*τ*_avg_ (ns) (at 535 nm)
1.0	—	535	0.4	0.85
1.0	TPE 0.2	446, 535	0.7	1.78
1.0	TPE 0.4	446, 536	1.2	1.86
1.0	TPE 0.6	446, 536	1.2	2.08
1.0	BrTPE 0.2	471, 535	5.9	5.16
1.0	BrTPE 0.4	472, 535	9.6	5.29
1.0	BrTPE 0.6	472, 536	13.8	5.55
1.0	TeBrTPE 0.2	475, 534	12.2	4.04
1.0	TeBrTPE 0.4	475, 535	16.2	4.25
1.0	TeBrTPE 0.6	464, 535	18.6	4.57

## Data Availability

Not applicable.

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
