# Peer review of "Three-Color White Photoluminescence Emission Using Perovskite Nanoplatelets and Organic Emitter"

_molecules, 2022, doi:10.3390/molecules27133982_

Round 1

Reviewer 1 Report

The author changed the manuscript according my previous comments. I do not understand why the authors need to divide the PLQY by the I0 value.

Reviewer 2 Report

In this work, the author applied an organic light-emitting ligand with AIE characteristics to CsPbBr3, improved the luminous efficiency of perovskite light-emitting material and obtained a film that simultaneously emitted green and blue light. The interaction of the perovskite crystal domain as well as the PL intensity of CsPbBr3 were increased. As a result, upon deposition of a red phosphorescent material, CBP:Ir(piq)3, onto a CsPbBr3 film with a TeBrTPE/CsPbBr3 molar ratio of 0.4, which exhibited three-color white-light PL emission with a CIE value of (0.25, 0.36). This work is meaningful for promoting white perovskite LEDs. Thus, I think this manuscript is suitable for publication in Molecules journal after some revision as follows: 

1. The author should further carried out the device fabrication and measurement.

2. The use of abbreviations is confusing such as “THF” in page 3 and 6, PL quantum yield and PLQY in page 4 and 6. Please author to check it.

3. There are some grammar and spelling errors. The author check the whole text and suppress them.

4. Formats of reference 3, 7, 9 and 26 are inconsistent with the others. Please author to check it.

Author Response

This manuscript is a resubmission of an earlier submission. The following is a list of the peer review reports and author responses from that submission.

Round 1

Reviewer 1 Report

AMX3-based metal halide perovskites have been widely studied in light-emitting diodes, optoelectronic devices, and other fields due to their excellent color purity, charge carrier mobility, and tunable optical bandgap. However, currently obtained perovskite emitters by mixing with organic molecules can only achieve two-color white-light emission based on blue and yellow. In this manuscript, the authors prepared three TPE derivatives with different numbers of bromide groups. By applying these AIE molecules to CsPbBr3 at different molar ratios, films emitting both green and blue light were obtained. Among them, the authors found that TeBrTPE molecules with four sites replaced by Br showed a strong AIE effect. Moreover, when TeBrTPE molecules were added to the CsPbBr3 film at a molar ratio of 0.6, the photoluminescence quantum yield value of CsPbBr3 was about 47 times higher than that of pure CsPbBr3. Finally, after depositing the red phosphorescent material CBP:Ir(piq)3 onto the CsPbBr3 film with a TeBrTPE/CsPbBr3 molar ratio of 0.4, three different wavelengths of 469, 535 ,613 nm and white light emission were observed. The work is effective, and can be recommended to be accepted after finishing the following minor revisions:

1. In the abstract, page 1, line 19, change “has” to “had”.

2. In the main text, page 2, line 48, change “phenylethylammoniu, (PEA+)” to “phenylethylammoniu (PEA+),”.

3. In the main text, page 3, line 112, change “and brine” to “brine”.

4. In the main text, page 4, line 157, change “exhibit” to “exhibited”.

5. In the main text, page 5, line 220, change “is” to “was”.

6. In the main text, page 9, line 323, change “improves” to “improved”.

7. In the main text, page 10, line 365, change “has” to “had”.

8. In the main text, page 10, line 372, change “is” to “was”.

9. In the conclusion, page 11, line 380, change “emits” to “emitted”.

10. Some related literature papers are recommended: Angew. Chem. Int. Ed. 2022. 61. e202116085; Aggregate. 2020. 1. 6-18; Adv. Mater. 2021. 33. 2105418; Aggregate 2022. DOI: org/10.1002/agt2.199; Angew. Chem. Int. Ed. 2021. 60. 15724.

11. In the main text, Figure 2, Figure 3, and Figure 4 only show the fluorescence spectra of TPE derivatives or perovskite films. The corresponding fluorescence photographs need to be added to make the experimental results more intuitive.

Reviewer 2 Report

The authors fabricate an organic–inorganic mixed film with three-color white EL emission by combining perovskite together with AIE materials. I recommend acceptance of this work if the following points were resolved.

1.     EQE (External quantum efficiency) is usually used to evaluate device efficiency. Please provide the EQE value of this white perovskite LED and is better to make a comparison to other reported perovskite LEDs.  

2       In addition to 1HNMR, 13CNMR and mass spectrometry are also necessary for the synthesis molecule characterization

Reviewer 3 Report

The authors present a work about the improvement of the luminous efficiency of the perovskite light-emitting material by applying organic light-emitting ligand with AIE characteristics.

The authors compare the PL intensity of different samples, which is not a valid method since the intensity of the spectra is arbitrary.

I also would recommend to use some technique that would allow to get more information about the aggregation of the different solution.

So, I only can accept the manuscript after an extensive improvement according to the following suggestions:

I recommend removing the abbreviature “AIE” from the title.

It is necessary to add more detail about the experimental setup used for the PL measurements: what was the excitation source? Were the spectra corrected for the spectral response of the equipment?

The supplier of the chemicals indicated on lines 89-103 is not indicated.

In line 162, I would replace statements like this “Because TPE has a large bandgap”, by a less subjective one. Can authors indicate the band gap value?

What are the units of the spectra shown in Figure 2a? I guess they are arbitrary units. This must be indicated. I suggest normalizing the spectra for better comparison.

In Figure 2b the authors present the units I/I0. What that means? The error value must be added.

The authors compare the intensity of different solutions. This can not be done. PL is not a quantitative technique. If the authors want to compare different solutions, they must use an absolute measurement such as the emission quantum yield value. As they do next.

In Figure 4, the PL intensity is in arbitrary units. The intensity of different samples cannot be compared.

In Table 1 the error of the PLQY and lifetime values must be added. Also must be indicated the method used to get the lifetime value from the measured  decay curves.

The Figure 5 a, b and c is too small.

I also would suggest adding an optical picture of the sample of Figure 6 under irradiation.

Round 2

Reviewer 3 Report

The author corrected the manuscript according almost all my comments. Despite that they still compare the intensity of different solutions. Since I think that is not a valid method, I can not agree with the publication.